# Intrinsic Information-Theoretic Models

**DOI:** 10.3390/e26050370

**Published:** 2024-04-28

**Authors:** D. Bernal-Casas, J. M. Oller

**Affiliations:** Department of Genetics, Microbiology and Statistics, Faculty of Biology, Universitat de Barcelona, 08028 Barcelona, Spain

**Keywords:** information geometry, fisher’s information, riemannian manifolds, schrödinger’s equation, principle of minimum fisher’s information, quantum harmonic oscillator, bayes’ theorem

## Abstract

With this follow-up paper, we continue developing a mathematical framework based on information geometry for representing physical objects. The long-term goal is to lay down informational foundations for physics, especially quantum physics. We assume that we can now model information sources as univariate normal probability distributions N (μ, $\sigma_{0})$, as before, but with a constant $\sigma_{0}$ not necessarily equal to 1. Then, we also relaxed the independence condition when modeling *m* sources of information. Now, we model *m* sources with a multivariate normal probability distribution $N_{m}\left(\mu ,\Sigma_{0}\right)$ with a constant variance–covariance matrix $\Sigma_{0}$ not necessarily diagonal, i.e., with covariance values different to 0, which leads to the concept of modes rather than sources. Invoking Schrödinger’s equation, we can still break the information into *m* quantum harmonic oscillators, one for each mode, and with energy levels independent of the values of $\sigma_{0}$, altogether leading to the concept of “intrinsic”. Similarly, as in our previous work with the estimator’s variance, we found that the expectation of the quadratic Mahalanobis distance to the sample mean equals the energy levels of the quantum harmonic oscillator, being the minimum quadratic Mahalanobis distance at the minimum energy level of the oscillator and reaching the “intrinsic” Cramér–Rao lower bound at the lowest energy level. Also, we demonstrate that the global probability density function of the collective mode of a set of *m* quantum harmonic oscillators at the lowest energy level still equals the posterior probability distribution calculated using Bayes’ theorem from the sources of information for all data values, taking as a prior the Riemannian volume of the informative metric. While these new assumptions certainly add complexity to the mathematical framework, the results proven are invariant under transformations, leading to the concept of “intrinsic” information-theoretic models, which are essential for developing physics.

## 1. Introduction

In this work, we continue developing the mathematical framework introduced in [1] by implementing some variations to better account for reality. In particular, we model information sources as univariate normal probability distributions N (μ, $\sigma_{0})$ as before but with a constant $\sigma_{0}$ not necessarily equal to 1. We also relax the independence condition when modeling *m* sources of information. Thus, we model *m*-dependent sources with a multivariate normal probability distribution $N_{m}$ (μ, $\Sigma_{0})$ with a constant variance–covariance matrix $\Sigma_{0}$ not necessarily diagonal, i.e., with covariance values different than 0, which leads to the concept of modes rather than sources when finding the solutions.

As in our initial work, the mathematical approach departs from the supposition that physical objects are information-theoretic in origin, an idea that has recurrently appeared in physics. In the following mathematical developments, we discover that the approach is importantly “intrinsic”, giving rise to the paper’s title, which is the main feature we want to emphasize in this study. In other words, regardless of how we parametrize the modeling, the approach’s inherent properties, for example, energy levels, remain the same, irrespective of updating the framework with the above modifications.

This entire work builds upon this finding, which makes it ideal for studying the properties of information representation and developing physics, and our modifications can further improve the framework’s accuracy and applicability to real-world scenarios. The long-term goal is to provide models to explain the “pre-physics” stage from which everything may emerge. We refer to the initial preprocessing of the source data information which is performed, in principle, by our sensory systems or organs. Therefore, the research in this follow-up paper may significantly contribute to the field and potentially guide future work in the area.

## 2. Mathematical Framework

The plan of this section, which we divide into eleven subsections for didactic purposes, is the following. In Section 2.1, we outline modeling a single source with a single sample and the derivation of Fisher’s information and the Riemannian manifold. In Section 2.2, we describe modeling a single source with *n* samples. Section 2.3 is devoted to analyzing the stationary states of a single source with *n* samples in the Riemannian manifold. In Section 2.4, we present the solutions of the stationary states in our formalism. In Section 2.5, we compute the probability density function, the mean quadratic Mahalanobis distance, and the “intrinsic” Cramér–Rao lower bound for a single source with *n* samples. An extension of this approach to *m* independent sources is conducted in Section 2.6 to compute the global probability density function at the ground-state level. In Section 2.7, we outline modeling *m*-dependent sources of a single sample, Fisher’s information, and the Riemannian manifold. Section 2.8 describes *m*-dependent sources of *n* samples. In Section 2.9, we analyze the stationary states of *m*-dependent sources of *n* samples in the Riemannian manifold. Section 2.10 is devoted to finding the solutions. Finally, in Section 2.11, we use Bayes’ theorem to obtain the posterior probability density function.

### 2.1. A Single Source with a Single Sample: The Fisher’s Information, the Riemannian Manifold, and the Quadratic Mahalanobis Distance

We start our mathematical description by modeling a single source with a univariate normal probability distribution $N(\mu ,\sigma_{0})$ where $\sigma_{0}>0$ is a known constant. This is a well-known parametric statistical model in which unidimensional parameter space may be identified with the real line, i.e., Θ=R. We can compute all the relevant quantities relevant to our purpose. For a single sample, the univariate normal density (with respect to the Lebesgue measure), its natural logarithm, and the partial derivative with respect to the parameter μ are given by
(1)$$\begin{array}{ccc} f(x;\mu ) & = & \frac{1}{\sqrt{2\pi }\sigma_{0}}e^{-\frac{1}{2\sigma_{0}^{2}}{\left(x-\mu \right)}^{2}}\;, \end{array}$$
(2)$$\begin{array}{ccc} \ln f & = & -\frac{1}{2}\ln\left(2\pi \sigma_{0}^{2}\right)-\frac{1}{2\sigma_{0}^{2}}{\left(x-\mu \right)}^{2}\;, \end{array}$$
(3)$$\begin{array}{ccc} \frac{\partial \ln f}{\partial \mu } & = & \frac{x-\mu }{\sigma_{0}^{2}}\;. \end{array}$$
From Equation (3), which is also called the score function, we can calculate Fisher’s information [2] for a single sample as
(4)$$I\left(\mu \right)=E_{\mu }\left({\left(\frac{\partial \ln f}{\partial \mu }\right)}^{2}\right)=E_{\mu }\left({\left(\frac{x-\mu }{\sigma_{0}^{2}}\right)}^{2}\right)=\frac{1}{\sigma_{0}^{2}}\;E_{\mu }\left({\left(\frac{x-\mu }{\sigma_{0}}\right)}^{2}\right)=\frac{1}{\sigma_{0}^{2}}\;,$$
since with the change $z=\frac{x-\mu }{\sigma_{0}}$, we have that $E_{\mu }\left({\left(\frac{x-\mu }{\sigma_{0}}\right)}^{2}\right)=\int_{-\infty }^{\infty }z^{2}\frac{1}{\sqrt{2\pi }}e^{-\frac{1}{2}z^{2}}dz=1$.

The Riemannian metric [3] from a single source with a single sample derived from the Fisher’s information (Equation 4) is a metric tensor whose covariant component, contravariant component, and determinant, respectively, are
(5)$$\begin{array}{ccc} g_{11}\left(\mu \right) & = & \frac{1}{\sigma_{0}^{2}}\;, \end{array}$$
(6)$$\begin{array}{ccc} g^{11}\left(\mu \right) & = & \sigma_{0}^{2}\;, \end{array}$$
(7)$$\begin{array}{ccc} \det(g(\mu )) & = & \frac{1}{\sigma_{0}^{2}}\;. \end{array}$$

The corresponding square of the Riemannian distance induced in the parametric space is the well-known quadratic Mahalanobis distance [4], i.e.,
(8)$$d_{M}^{2}\left(\mu^{2},\mu^{1}\right)=\frac{1}{\sigma_{0}^{2}}{\left(\mu^{2}-\mu^{1}\right)}^{2}\;.$$
The quadratic Mahalanobis distance will play a critical role in the next sections.

### 2.2. A Single Source with *n* Samples: The Fisher’s Information, the Riemannian Manifold, and the Square of the Riemannian Distance

If the source generates *n* samples, $\left(x^{1},\cdots ,x^{n}\right)$, drawn independently from a univariate normal probability distribution (Equation 1), the likelihood of this *n* variate sample, its log-likelihood, and the scores, respectively, are
(9)$$\begin{array}{ccc} f_{n}\left(x^{1},\cdots ,x^{n};\mu \right) & = & \prod_{i=1}^{n}f\left(x^{i};\mu \right)={\left(\frac{1}{2\pi \sigma_{0}^{2}}\right)}^{\frac{n}{2}}\exp\left(-\frac{1}{2\sigma_{0}^{2}}\sum_{i=1}^{n}{\left(x^{i}-\mu \right)}^{2}\right)\;, \end{array}$$
(10)$$\begin{array}{ccc} \ln f_{n} & = & -\frac{n}{2}\ln\left(2\pi \sigma_{0}^{2}\right)-\frac{1}{2\sigma_{0}^{2}}\sum_{i=1}^{n}{\left(x^{i}-\mu \right)}^{2}\;, \end{array}$$
(11)$$\begin{array}{ccc} \frac{\partial \ln f_{n}}{\partial \mu } & = & \frac{1}{\sigma_{0}^{2}}\;\sum_{i=1}^{n}\left(x^{i}-\mu \right)=\frac{n}{\sigma_{0}^{2}}\left(\bar{x}-\mu \right)\;. \end{array}$$
Likewise, from Equation (11) we can calculate the Fisher’s information corresponding to an *n* size sample as
(12)$$I_{n}\left(\mu \right)=E_{\mu }\left({\left(\frac{\partial \ln f_{n}}{\partial \mu }\right)}^{2}\right)=\frac{n}{\sigma_{0}^{2}}\;E_{\mu }\left({\left(\frac{\left(\bar{x}-\mu \right)}{\sigma_{0}/\sqrt{n}}\right)}^{2}\right)=\frac{n}{\sigma_{0}^{2}}\;.$$
since $\bar{x}$ follows a univariate normal distribution with mean equal to μ and variance equal to $\frac{\sigma_{0}^{2}}{n}$.

In other words
(13)$$I_{n}\left(\mu \right)=nI\left(\mu \right)\;,$$
which shows the well-known additive property of the Fisher information for independent samples.

The Riemannian metric [3] from a single source with *n* samples derived from the Fisher’s information (Equation 12) is a metric tensor whose covariant component, contravariant component, and determinant, respectively, are
(14)$$\begin{array}{ccc} {\tilde{g}}_{11}\left(\mu \right) & = & \frac{n}{\sigma_{0}^{2}}\;, \end{array}$$
(15)$$\begin{array}{ccc} {\tilde{g}}^{11}\left(\mu \right) & = & \frac{\sigma_{0}^{2}}{n}\;, \end{array}$$
(16)$$\begin{array}{ccc} \det(\tilde{g}\left(\mu \right)) & = & \frac{n}{\sigma_{0}^{2}}\;. \end{array}$$

The square of the Riemannian distance, $\rho^{2}$, induced by the Fisher information matrix corresponding to a sample of arbitrary size *n* will be equal to *n* times the quadratic Mahalanobis distance (Equation 8), i.e.,
(17)$$\rho^{2}\left(\mu^{2},\mu^{1}\right)=n\;d_{M}^{2}\left(\mu^{2},\mu^{1}\right)=\frac{n}{\sigma_{0}^{2}}{\left(\mu^{2}-\mu^{1}\right)}^{2}\;.$$

### 2.3. Stationary States of a Single Source of n Samples in the Riemannian Manifold

To calculate the stationary states of a single source of *n* samples, we can invoke the principle of minimum Fisher information [5] or use the time-independent non-relativistic Schrodinger’s equation [6]. The two approaches have been demonstrated to be equivalent elsewhere [5]. The equation reads as follows
(18)$$-k\nabla^{2}\psi \left(\mu \right)+U\left(\mu \right)\psi \left(\mu \right)=\lambda \psi \left(\mu \right)\;,$$
where U(μ) is a potential energy and k,λ>0. The solution must also satisfy $\lim_{\mu \to -\infty }\psi \left(\mu \right)=\lim_{\mu \to +\infty }\psi \left(\mu \right)=0$ and $\int_{-\infty }^{\infty }\psi^{2}\left(\mu \right)d\mu =1$. For simplicity, we will write ψ instead of ψ(μ).

We can use the modulus square of the score function (11) as the potential energy, except for a constant term
(19a)∂lnfn∂μ2=g11(μ)∂lnfn∂μ2,(19b)=σ02nnσ02x¯−μ2,(19c)=nσ02x¯−μ2.

Alternatively, we can use as the potential energy the difference between the maximum of the log-likelihood attained by the sample, $\left(x^{1},\ldots ,x^{n}\right)$, minus the log-likelihood at an arbitrary point μ, up to a proportionality constant. Since the likelihood is given by (Equation 9), we can rewrite it as
(20)$$f_{n}\left(x^{1},\cdots ,x^{n};\mu \right)={\left(\frac{1}{2\pi \sigma_{0}^{2}}\right)}^{\frac{n}{2}}\exp\left(-\frac{n}{2\sigma_{0}^{2}}\;s_{n}^{2}\right)\exp\left(-\frac{n}{2\sigma_{0}^{2}}\;{\left(\bar{x}-\mu \right)}^{2}\right)\;,$$
where $\bar{x}=\frac{1}{n}\sum_{i=1}^{n}x_{i}$ and $s_{n}^{2}=\frac{1}{n}\sum_{i=1}^{n}{\left(x_{i}-\bar{x}\right)}^{2}$. The supreme likelihood is obviously attained when $\mu =\bar{x}$, then, the previously mentioned potential will be
(21)$$U\left(\mu \right)\propto \ln f_{n}\left(x^{1},\cdots ,x^{n};\bar{x}\right)-\ln f_{n}\left(x^{1},\cdots ,x^{n};\mu \right)=\frac{n}{2\sigma_{0}^{2}}\;{\left(\bar{x}-\mu \right)}^{2}\;.$$
This expression is up to a proportionality constant equal to (19). Thus, we may choose as the potential energy $U\left(\mu \right)=\frac{nC}{\sigma_{0}^{2}}{\left(\bar{x}-\mu \right)}^{2}$ with C>0, and Equation (Equation 18) reads as:(22)$$-k\nabla^{2}\psi +\frac{nC}{\sigma_{0}^{2}}{\left(\bar{x}-\mu \right)}^{2}\psi =\lambda \psi \;,$$
We compute the Laplacian in Equation (Equation 22) as:(23a)∇2ψ=1|g(μ)|∂∂μ|g(μ)|g11(μ)∂ψ∂μ,(23b)=σ0n∂∂μnσ0σ02n∂ψ∂μ,(23c)=σ02n∂2ψ∂μ2=σ02nψ′′.
Inserting Equation (Equation 23) into Equation (Equation 22), we obtain:(24)$$-\frac{k\sigma_{0}^{2}}{n}\psi^{\prime \prime }+\frac{nC}{\sigma_{0}^{2}}{\left(\bar{x}-\mu \right)}^{2}\psi =\lambda \psi \;,$$
which is Schrödinger’s equation of the quantum harmonic oscillator [7].

### 2.4. Solutions of a Single Quantum Harmonic Oscillator in the Riemannian Manifold

Some steps now may seem obvious for those used to quantum mechanics. Considering that ψ has the following form:(25)$$\psi \left(\mu \right)=\gamma e^{\eta }\;,\;\;\;\;\mathrm{with}\;\gamma >0\;\mathrm{real}\;,\;\eta \;a\;\mathrm{function}\;\mathrm{of}\;\mu \;.$$
Equation (Equation 24) results:(26a)−kσ02nγeη(η′)2+γeηη′′+nCσ02x¯−μ2γeη=λγeη,(26b)−kσ02n(η′)2+η′′+nCσ02x¯−μ2=λ.
Assuming a solution for η(μ) with the form
(27)$$\eta \left(\mu \right)=-\xi n{\left(\bar{x}-\mu \right)}^{2}\;,\;\;\;\;\mathrm{with}\;\xi >0\;,$$
And inserting this expression into Equation (Equation 26) gives
(28)$$-\frac{k\;\sigma_{0}^{2}}{n}\left(4\xi^{2}n^{2}{\left(\bar{x}-\mu \right)}^{2}-2\xi n\right)+\frac{nC}{\sigma_{0}^{2}}{\left(\bar{x}-\mu \right)}^{2}=\lambda \;,$$
which implies that
(29)$$4k\sigma_{0}^{2}\xi^{2}=\frac{C}{\sigma_{0}^{2}}\;,\;\;\;\;2k\sigma_{0}^{2}\xi =\lambda \;.$$
In other words, k,C,λ,ξ can not be chosen arbitrarily because they have to satisfy these equations. For example, we can choose $k=\frac{2}{n}$ and $C=\frac{1}{2n}$, which forces $\xi =\frac{1}{4\sigma_{0}^{2}}$, and $\lambda =\frac{1}{n}$. Therefore, we can write
(30)$$-\frac{2\sigma_{0}^{2}}{n^{2}}\psi^{\prime \prime }+\frac{1}{2\sigma_{0}^{2}}{\left(\bar{x}-\mu \right)}^{2}\psi =\frac{1}{n}\psi \;,$$
whose solution is given by
(31)$$\psi \left(\mu \right)=\gamma e^{-\frac{1}{4\sigma_{0}^{2}}n{\left(\bar{x}-\mu \right)}^{2}}\;.$$
With this configuration, we compute the normalization constant γ
(32)$$1=\int_{-\infty }^{\infty }\psi^{2}\left(\mu \right)d\mu =\gamma^{2}\int_{-\infty }^{\infty }e^{-\frac{1}{2\sigma_{0}^{2}}n{\left(\bar{x}-\mu \right)}^{2}}d\mu =2\gamma^{2}\sigma_{0}\int_{0}^{\infty }e^{-\frac{1}{2}nt^{2}}dt\;,$$
where we used a first change of variable $\frac{\bar{x}-\mu }{\sigma_{0}}=t$. Now, using a second change of variable $\frac{1}{2}nt^{2}=s$, $dt=\sqrt{\frac{2}{n}}\frac{1}{2}s^{-\frac{1}{2}}ds$, Equation (Equation 32) writes as
(33)$$1=2\gamma^{2}\sigma_{0}\int_{0}^{\infty }e^{-s}\sqrt{\frac{2}{n}}\frac{1}{2}s^{-\frac{1}{2}}ds=\gamma^{2}\sqrt{\frac{2}{n}}\sigma_{0}\int_{0}^{\infty }s^{-\frac{1}{2}}e^{-s}ds=\gamma^{2}\sqrt{\frac{2}{n}}\sqrt{\pi }\sigma_{0}\;.$$

Isolating γ from Equation (Equation 33), we obtain $\gamma ={\left(\frac{n}{2\pi \sigma_{0}^{2}}\right)}^{\frac{1}{4}}$. Therefore, Equation (Equation 31) reads as
(34)$$\psi \left(\mu \right)={\left(\frac{n}{2\pi \sigma_{0}^{2}}\right)}^{\frac{1}{4}}e^{-\frac{1}{4\sigma_{0}^{2}}n{\left(\bar{x}-\mu \right)}^{2}}=\psi_{0}\left(\mu \right)\;,$$
which is the ground-state solution of the quantum harmonic oscillator problem, and the wave function for the ground-state. The solutions of the quantum harmonic oscillator involve Hermite Polynomials, which were introduced elsewhere [8,9]. In this way, we can prove, after some tedious but straightforward computations, that the wave function:(35)$$\psi_{1}\left(\mu \right)=\gamma_{1}\left(\frac{\bar{x}-\mu }{\sigma_{0}}\right)e^{-\frac{1}{4\sigma_{0}^{2}}n{\left(\bar{x}-\mu \right)}^{2}}\;,\;\;\;\;\mathrm{with}\;\gamma_{1}>0\;,$$
is also a solution of
(36)$$-\frac{2\sigma_{0}^{2}}{n^{2}}\psi^{\prime \prime }+\frac{1}{2\sigma_{0}^{2}}{\left(\bar{x}-\mu \right)}^{2}\psi =\lambda_{1}\psi_{1}\;,$$
where $\lambda_{1}=\frac{3}{n}$ is the energy of the first excited state, and $\gamma_{1}={\left(\frac{n^{3}}{2\pi \sigma_{0}^{2}}\right)}^{\frac{1}{4}}$ is the normalization constant. With this representation, the λ’s (energy levels) are given by
(37)$$\lambda_{\nu }=\frac{2}{n}\left(\nu +\frac{1}{2}\right)=E_{\nu }\;,\;\;\;\;\;\mathrm{with}\;\nu =0,1,\cdots \;.$$
Looking closely at Equation (Equation 37), we appreciate that the energy levels depend on two numbers, ν and *n*. The ground state at ν=0 has a finite energy $E_{0}=\frac{1}{n}$, and can become arbitrarily close to zero by massive sampling. Notably, the energy levels are independent of $\sigma_{0}$. In other words, they do not depend on the informative parameters, leading to the concept of “intrinsic” information-theoretic models which will be discussed in greater detail later.

### 2.5. Probability Density Function of a Single Source of n Samples, Mean Quadratic Mahalanobis Distance, and Intrinsic Cramér–Rao Lower Bound

Assuming that the square modulus of the wave function can be interpreted as the probability density function:(38)$${∥\psi \left(\mu \right)∥}^{2}=\psi^{*}\left(\mu \right)\psi \left(\mu \right)=\rho \left(\mu \right)\;,$$
we can compute the performance of the estimations of μ. For instance, we can calculate the expectation of the quadratic Mahalanobis distance (Equation 8) to the sample mean $\bar{x}$ at the ground state (Equation 34), obtaining
(39)$$E_{\mu ,\rho_{0}\left(\mu \right)}\left({\left(\frac{\bar{x}-\mu }{\sigma_{0}}\right)}^{2}\right)=\frac{1}{n}\int_{-\infty }^{\infty }{\left(\frac{\bar{x}-\mu }{\sigma_{0}/\sqrt{n}}\right)}^{2}\frac{1}{\sqrt{2\pi {\left(\sigma_{0}/\sqrt{n}\right)}^{2}}}e^{-\frac{1}{2{\left(\sigma_{0}/\sqrt{n}\right)}^{2}}{\left(\bar{x}-\mu \right)}^{2}}d\mu =\frac{1}{n}\;.$$
Likewise, we can compute the expectation of the quadratic Mahalanobis distance (Equation 8) to the sample mean $\bar{x}$ at the first excited state, obtaining
(40)$$E_{\mu ,\rho_{1}\left(\mu \right)}\left({\left(\frac{\bar{x}-\mu }{\sigma_{0}}\right)}^{2}\right)=\frac{1}{n}\int_{-\infty }^{\infty }\frac{1}{\sqrt{2\pi {\left(\sigma_{0}/\sqrt{n}\right)}^{2}}}{\left(\frac{\bar{x}-\mu }{\sigma_{0}/\sqrt{n}}\right)}^{4}e^{-\frac{1}{2{\left(\sigma_{0}/\sqrt{n}\right)}^{2}}{\left(\bar{x}-\mu \right)}^{2}}d\mu =\frac{3}{n}\;.$$
The expectation of the quadratic Mahalanobis distances to the sample mean $\bar{x}$ at the different states equal the quantum harmonic oscillator’s energy levels, i.e., this quantity is definitively quantized. Interestingly, the expectation of the quadratic Mahalanobis distance to the sample mean $\bar{x}$ at the ground state (Equation 39) equals the intrinsic Cramér–Rao lower bound (ICRLB) for unbiased estimators
(41)$$E_{\mu }\left({\left(\frac{\bar{x}-\mu }{\sigma_{0}}\right)}^{2}\right)\geq \frac{m}{n}{|}_{m=1}=\frac{1}{n}\;,$$
considering that we are modeling a single source of *n* samples with a single informative parameter μ, i.e., m=1. For further details, see [10].

### 2.6. m Independent Sources of *n* Samples and Global Probability Density Function

With *m* independent sources, each generating *n* samples, a finite set of *m* quantum harmonic oscillators may represent reality. Presuming independence of the sources of information, the “global” wave function (also called the collective wave function) can factor as the product of single wave functions. We can write the global wave function as
(42)$$\psi \left(\mu \right)=\psi \left(\mu^{1},\mu^{2},\cdots ,\mu^{m}\right)=\psi \left(\mu^{1}\right)\psi \left(\mu^{2}\right)\cdots \psi \left(\mu^{m}\right)\;,$$
It constitutes a many-body system, and we may refer to the vector μ as the μ field.

For example, in the case of modeling two independent sources, the global wave function at the ground state will be the product of single wave functions, each of them at the ground state
(43a)ψ0(μ)=ψ0(μ1)ψ0(μ2),(43b)=n2πσ0214exp{−14σ02nx¯1−μ12}n2πσ0214exp{−14σ02nx¯2−μ22},(43c)=n2πσ0214n2πσ0214exp{−14σ02nx¯1−μ12+nx¯2−μ22},(43d)=n2πσ0212exp{−14σ02nx¯1−μ1,x¯2−μ2x¯1−μ1x¯2−μ2}.
We can generalize Equation (Equation 43) for having *m* independent sources. The global wave function is written as
(44)$$\psi_{0}\left(\mu \right)={\left(\frac{n}{2\pi \sigma_{0}^{2}}\right)}^{\frac{m}{4}}\exp\{-\frac{1}{4\sigma_{0}^{2}}\;n\;{\left(\bar{x}-\mu \right)}^{T}\left(\bar{x}-\mu \right)\}\;.$$
Using Equation (Equation 38), the probability density function is:(45)$$\rho_{0}\left(\mu \right)={\left(\frac{n}{2\pi \sigma_{0}^{2}}\right)}^{\frac{m}{2}}\exp\{-\frac{1}{2\sigma_{0}^{2}}\;n\;{\left(\bar{x}-\mu \right)}^{T}\left(\bar{x}-\mu \right)\}\;.$$

### 2.7. m Dependent Sources of a Single Sample, the Fisher’s Information, the Riemannian Manifold, and the Quadratic Mahalanobis Distance

Consider now *m* possibly dependent sources, generating a multivariate sample of size 1, x. Now, although we have a finite set of *m* univariate quantum harmonic oscillators that can also represent reality, since these sources may be dependent, it is convenient to model this situation as a single *m* variate source with an *m*-variate random vector x following a *m* variate normal probability distribution $N_{m}\left(\mu ,\Sigma_{0}\right)$ where $\mu \in R^{m}$ and $\Sigma_{0}$ is a known constant strictly positive definite m×m matrix, the covariance matrix of random vector x, i.e., $cov\left(x\right)=\Sigma_{0}>0$. This is a well-known parametric statistical model in which their *m*-dimensional parameter space may be identified with $\Theta =R^{m}$; for further details, see [11]. Identifying, as is customary, the points of the manifold Θ with their coordinates $\mu =(\mu^{1},\ldots ,\mu^{m})$, we can compute all the quantities relevant to our purpose. For a single sample, the *m*-variate normal density (with respect to the Lebesgue measure), its natural logarithm, and the partial derivative with respect to $\mu^{i}$ are given by
(46)$$\begin{array}{ccc} f_{m}\left(x;\mu \right) & = & {\left(2\pi \right)}^{-\frac{m}{2}}\det{\left(\Sigma_{0}\right)}^{-\frac{1}{2}}\exp\left(-\frac{1}{2}{\left(x-\mu \right)}^{T}\Sigma_{0}^{-1}\left(x-\mu \right)\right)\;, \end{array}$$
(47)$$\begin{array}{ccc} \ln f_{m} & = & -\frac{m}{2}\ln\left(2\pi \right)-\frac{1}{2}\ln\left(\det\left(\Sigma_{0}\right)\right)-\frac{1}{2}{\left(x-\mu \right)}^{T}\Sigma_{0}^{-1}\left(x-\mu \right)\;, \end{array}$$
(48)$$\begin{array}{ccc} \frac{\partial \ln f_{m}}{\partial \mu^{i}} & = & \sum_{\alpha =1}^{m}\sigma^{i\alpha }\left(x^{\alpha }-\mu^{\alpha }\right)\;. \end{array}$$
where, following a standard matrix calculus notation, $\sigma^{i\alpha }$ is the element located in row *i* and column α of the inverse covariance matrix $\Sigma_{0}$.

The Fisher’s information matrix $G=g_{ij}\left(\mu \right)$ is a m×m matrix whose elements are
(49a)gij(μ)=Eμ∂lnfm∂μi∂lnfm∂μj,(49b)=Eμ∑α=1mσiα(xα−μα)∑β=1mσjβ(xβ−μβ,(49c)=∑α=1m∑β=1mσiασjβEμ(xα−μα)(xβ−μβ),(49d)=∑β=1m∑α=1mσiασαβσβj=∑β=1mδβiσβj=σij,
where we have taken into account the symmetry of $\sigma^{j\beta }$, and that $cov\left(\left(x^{\alpha }-\mu^{\alpha }\right),\left(x^{\beta }-\mu^{\beta }\right)\right)=E_{\mu }\left(\left(x^{\alpha }-\mu^{\alpha }\right)\left(x^{\beta }-\mu^{\beta }\right)\right)=\sigma_{\alpha \beta }$, or, in matrix form, $G=\Sigma_{0}^{-1}$.

It is well known that the Fisher information matrix is a second-order covariant tensor of the parameter space. It is positive definite and may be considered the metric tensor of this manifold, giving it the structure of the Riemannian manifold. To avoid confusion, we must emphasize that the subscripts or superscripts used to reference the variance and covariance matrix or its inverse do not have a tensor character, i.e., the components of the metric tensor $g_{ij}\left(\mu \right)$ are those of a covariant tensor, in tensor notation, they are written as subscripts and are equal to the components of the inverse variance–covariance matrix given in (Equation 49).

The Riemannian geometry induced by the Fisher information matrix in the parameter space is, in this case, Euclidean, and the square of the Riemannian distance, also known as the Rao distance, is the Mahalanobis distance given by
(50)$$d_{M}^{2}\left(\mu^{2},\mu^{1}\right)={\left(\mu^{2}-\mu^{1}\right)}^{T}\Sigma_{0}^{-1}\left(\mu^{2}-\mu^{1}\right)\;.$$
In this expression, the parameter space points are identified with their coordinates and written in matrix notation as m×1 column vectors.

All these results correspond to a multivariate with a sample size n=1.

### 2.8. m Dependent Sources of n Samples, the Fisher’s Information, the Riemannian Manifold, and the Square of the Riemannian Distance

If each of the *m* dependent sources generates *n* samples, $\left(x^{1},\cdots ,x^{n}\right)$, drawn independently from a multivariate normal probability distribution (Equation 46), the likelihood distribution is
(51a)fm,n(x1,⋯,xn;μ)=∏i=1n(2π)−m2det(Σ0)−12exp(−12(xi−μ)TΣ0−1(xi−μ)),(51b)=(2π)−mn2det(Σ0)−n2exp(−12∑i=1n(xi−μ)TΣ0−1(xi−μ)).

The summation term within the exponential function can be decomposed into two terms
(52a)∑i=1n(xi−μ)TΣ0−1(xi−μ)=∑i=1nTr(Σ0−1(xi−μ)(xi−μ)T),(52b)=TrΣ0−1∑i=1n(xi−μ)(xi−μ)T,(52c)=TrΣ0−1∑i=1n(xi−x¯+x¯−μ)(xi−x¯+x¯−μ)T,(52d)=TrΣ0−1∑i=1n(xi−x¯)(xi−x¯)T+∑i=1n(x¯−μ)(x¯−μ)T,(52e)=nTrΣ0−1Sn+ndM2(x¯,μ),
where $\bar{x}=\left(\sum_{i=1}^{n}x^{i}\right)/n$, and $S_{n}=\left(\sum_{i=1}^{n}\left(x^{i}-\bar{x}\right){\left(x^{i}-\bar{x}\right)}^{T}\right)/n$, i.e., the sample mean and the sample covariance matrix corresponding to this random sample is the quadratic Mahalanobis distance to the sample mean, $\bar{x}$.

Inserting Equation (Equation 52) into Equation (Equation 51) results
(53)$$f_{m,n}\left(x^{1},\cdots ,x^{n};\mu \right)={\left(2\pi \right)}^{-\frac{mn}{2}}\det{\left(\Sigma_{0}\right)}^{-\frac{n}{2}}\exp\left(-\frac{n}{2}\mathrm{Tr}\{\Sigma_{0}^{-1}S_{n}\}\right)\exp\left(-\frac{n}{2}d_{M}^{2}\left(\bar{x},\mu \right)\right)\;.$$

The log-likelihood distribution is
(54)$$\ln f_{m,n}=-\frac{mn}{2}\ln\left(2\pi \right)-\frac{n}{2}\ln\det\left(\Sigma_{0}\right)-\frac{n}{2}\mathrm{Tr}\{\Sigma_{0}^{-1}S_{n}\}-\frac{n}{2}d_{M}^{2}\left(\bar{x},\mu \right)\;,$$
and the partial derivative of $\ln f_{m,n}$ with respect to $\mu^{\alpha }$ using standard classical notation for covariant derivatives and additionally using repeated index summation convention is
(55)$$\frac{\partial \ln f_{m,n}}{\partial \mu^{\alpha }}={\left(\ln f_{m,n}\right)}_{,\;\alpha }=n\;g_{\alpha \beta }\left({\bar{x}}^{\beta }-\mu^{\beta }\right)\;.$$

The corresponding Fisher information matrix and the square of Riemannian distance for a sample of size *n* will be the above Equations (Equation 49) and (Equation 50) multiplied by *n*.

### 2.9. Stationary States of m-Dependent Sources of n Samples in the Riemannian Manifold

To calculate the stationary states, we can invoke the time-independent non-relativistic Schrodinger’s equation [6] as above. In the multivariate case, the wave equation reads as follows
(56)$$-k\nabla^{2}\psi \left(\mu \right)+U\left(\mu \right)\psi \left(\mu \right)=\lambda \psi \left(\mu \right)\;,$$
where U(μ) is the potential energy and k,λ>0. The solution must also satisfy that vanish to infinity and $\int_{R^{m}}\psi^{2}\left(\mu \right)d\mu =1$. For simplicity, we will write ψ instead of ψ(μ).

We can use the square of the norm of the log-likelihood gradient as the potential energy, except for a constant term. The components of the gradient of $\left(\ln f_{m,n}\right)$, a contravariant vector field, observing that the inverse of the metric tensor corresponding to a sample of size *n* is given by $\frac{1}{n}g^{\gamma \alpha }$ since $g^{\gamma \alpha }\;g_{\alpha \beta }=\delta_{\beta }^{\gamma }$ where $\delta_{\beta }^{\gamma }$ is the Kronecker delta’s, will be given in classical notation by
(57)$${\left(\nabla \ln f_{m,n}\right)}^{\gamma }={\left(\ln f_{m,n}\right)}_{,}^{\;\gamma }=\frac{1}{n}\;g^{\gamma \alpha }{\left(\ln f_{m,n}\right)}_{,\;\alpha }=\frac{1}{n}\;g^{\gamma \alpha }\;n\;g_{\alpha \beta }\left({\bar{x}}^{\beta }-\mu^{\beta }\right)=\left({\bar{x}}^{\gamma }-\mu^{\gamma }\right)\;,$$
and, therefore, the square of the norm of the log-likelihood gradient will be
(58)$${∥\nabla \ln f_{m,n}∥}^{2}=n\;g_{\gamma \beta }{\left(\ln f_{m,n}\right)}_{,}^{\;\gamma }\;{\left(\ln f_{m,n}\right)}_{,}^{\;\beta }=n\;g_{\gamma \beta }\left({\bar{x}}^{\gamma }-\mu^{\gamma }\right)\left({\bar{x}}^{\beta }-\mu^{\beta }\right)=n\;d_{M}^{2}\left(\bar{x},\mu \right)\;.$$

Alternatively, we can use the difference between the log-likelihood at an arbitrary point $\mu_{0}=\bar{x}$ minus the log-likelihood attained by the sample as the potential
(59)$$U\left(\mu \right)\propto \ln f_{m,n}\left(\bar{x}\right)-\ln f_{m,n}\left(\mu \right)=\frac{n}{2}d_{M}^{2}\left(\bar{x},\mu \right)\;,$$
which is up to a proportionality constant equal to (Equation 58). Thus, Equations (Equation 58) and (Equation 59) suggest to take as the potential energy $U\left(\mu \right)=nC\;d_{M}^{2}\left(\bar{x},\mu \right)$ with C>0. In this way, Equation (Equation 56) reads as
(60)$$-k\nabla^{2}\psi +nC\;d_{M}^{2}\left(\bar{x},\mu \right)\psi =\lambda \psi \;.$$
To proceed, we must compute the Laplacian in Equation (Equation 60). If g=det(G), with G defined in (Equation 49), i.e., the determinant corresponding to the tensor of the information metric for samples of size n=1, for a sample of arbitrary size *n* that determinant will be equal to $n^{m}g$. In this way, the Laplacian of a function ψ will be given by
(61)$$\nabla^{2}\psi =\frac{1}{\sqrt{n^{m}g}}\;\frac{\partial }{\partial \mu^{i}}\left(\sqrt{n^{m}g}\;\frac{1}{n}\;g^{ij}\;\frac{\partial \psi }{\partial \mu^{j}}\right)=\frac{1}{n}\;g^{ij}\;\frac{\partial^{2}\psi }{\partial \mu^{i}\partial \mu^{j}}\;,$$
where we have used repeated index summation convention. For further details about this formula, see, for instance, [12]. Notice that $g^{ij}$ equals the variance–covariance matrix $\Sigma_{0}$. Moreover, i m×m matrix $\Psi_{\mu }^{\prime \prime }={\left(\frac{\partial^{2}\psi }{\partial \mu^{i}\partial \mu^{j}}\right)}_{m\times m}$, which is the Hessian matrix of ψ under the coordinates $\mu =(\mu^{1},\ldots ,\mu^{m})$, Equation (Equation 61) can be written as
(62)$$\nabla^{2}\psi =\frac{1}{n}\mathrm{Tr}\left(\Sigma_{0}\Psi_{\mu }^{\prime \prime }\right)\;.$$
Inserting Equation (Equation 62) into (Equation 60), we obtain
(63)$$-\frac{k}{n}\mathrm{Tr}\;\left(\Sigma_{0}\Psi_{\mu }^{\prime \prime }\right)\;+nC\;d_{M}^{2}\left(\mu ,{\bar{x}}_{n}\right)\psi =\lambda \psi \;,$$
which is the Schrödinger’s equation of *m*-coupled quantum harmonic oscillators.

### 2.10. Solutions of m-Coupled Quantum Harmonic Oscillators in the Riemannian Manifold

We observe that both (Equation 58) and (Equation 61) are invariant under coordinate changes on $\Theta =R^{n}$. Therefore, Equation (Equation 63) will remain invariant under these changes, particularly linear ones.

Since $\Sigma_{0}$ is a symmetric m×m matrix which diagonalizes on an orthonormal basis, it can be written as $\Sigma_{0}=UDU^{T}$, where U is an orthogonal m×m matrix and D is a diagonal m×m matrix $D=diag(\gamma_{1}^{2},\ldots ,\gamma_{m}^{2})$, i=1,⋯,m, where $\gamma_{\alpha }>0$ are the eigenvalues of the square root of the variance–covariance matrix $\Sigma_{0}$.

Thus, by introducing the change of coordinates $\eta =U^{T}\mu$ and $\bar{y}=U^{T}\bar{x}$, the metric tensor becomes diagonal, i.e., $\hat{G}=D^{-1}$. Taking this coordinate change into account, the Equation (Equation 58) becomes
(64)$$n\;d_{M}^{2}\left(\bar{x},\mu \right)=n\;d_{M}^{2}\left(\bar{y},\eta \right)=\sum_{i=1}^{m}\frac{1}{\gamma_{i}^{2}}{\left({\bar{y}}^{i}-\eta^{i}\right)}^{2}\;.$$
Moreover, if we define the symmetric m×m matrix $\Psi_{\eta }^{\prime \prime }={\left(\frac{\partial^{2}\Psi }{\partial \eta^{i}\partial \eta^{j}}\right)}_{m\times m}$, which is the Hessian matrix of ψ under the new coordinates $\eta =(\eta^{1},\ldots ,\eta^{m})$, the Equation (Equation 61) becomes
(65)$$\nabla^{2}\psi =\frac{1}{n}\mathrm{Tr}\left(\Sigma_{0}\Psi_{\mu }^{\prime \prime }\right)=\frac{1}{n}\mathrm{Tr}\left(D\Psi_{\eta }^{\prime \prime }\right)\;.$$
Making use of Equation (Equation 64) and Equation (Equation 65) in Equation (Equation 63), we obtain
(66)$$-\frac{k}{n}\mathrm{Tr}\;\left(D\Psi_{\eta }^{\prime \prime }\right)\;+nC\;d_{M}^{2}\left(\bar{y},\eta \right)\psi =\lambda \;\psi \;,$$
which is the Schrödinger’s equation of the *m*-decoupled quantum harmonic oscillators. If we choose $k=\frac{2}{n}$ and $C=\frac{1}{2n}$, Equation (Equation 66) can be written as
(67)$$\sum_{i=1}^{m}\left(-\frac{2\gamma_{i}^{2}}{n^{2}}\frac{\partial^{2}\psi }{\partial {\left(\eta^{i}\right)}^{2}}+\;\frac{1}{2\gamma_{i}^{2}}{\left({\bar{y}}^{i}-\eta^{i}\right)}^{2}\psi \right)=\lambda \;\psi \;.$$
Additionally, if we define $\lambda =\sum_{i=1}^{m}\lambda_{\alpha }$ with $\lambda_{\alpha }>0$, we may write
(68)$$\sum_{i=1}^{m}\left(-\frac{2\gamma_{i}^{2}}{n^{2}}\frac{\partial^{2}\psi }{\partial {\left(\eta^{i}\right)}^{2}}+\;\frac{1}{2\gamma_{i}^{2}}{\left({\bar{y}}^{i}-\eta^{i}\right)}^{2}\psi -\lambda_{\alpha }\;\psi \right)=0\;.$$
Note, that if $\psi_{\alpha }$ is a non–trivial solution of
(69)$$-\frac{2\gamma_{\alpha }^{2}}{n^{2}}\frac{\partial^{2}\psi_{\alpha }}{\partial {\left(\eta^{\alpha }\right)}^{2}}+\;\frac{1}{2\gamma_{\alpha }^{2}}{\left({\bar{y}}^{\alpha }-\eta^{\alpha }\right)}^{2}\psi_{\alpha }=\lambda_{\alpha }\;\psi_{\alpha }\;,\;\alpha =1,\ldots ,m\;.$$
then, Equation (Equation 66) admits a solution of the form
(70)$$\psi \left(\eta^{1},\ldots ,\eta^{m}\right)=\prod_{\alpha =1}^{m}\psi_{\alpha }\left(\eta^{\alpha }\right)\;.$$
Each of the Equations in (Equation 69) admits infinite solutions for different values of $\lambda_{\alpha }$, as in our previous work [1]. More specifically, it admits solutions for
(71)$$\lambda_{\alpha ,\nu }=\frac{2}{n}\left(\nu +\frac{1}{2}\right)\;,\;\nu \in N\;.$$
In particular, for ν=0, we have $\lambda_{\alpha ,0}=\frac{1}{n}$ and the wave function for the ground-state is
(72)$$\psi_{\alpha ,0}\left(\eta^{\alpha }\right)={\left(\frac{n}{2\pi \gamma_{\alpha }^{2}}\right)}^{\frac{1}{4}}e^{-\frac{1}{4\gamma_{\alpha }^{2}}n{\left({\bar{y}}^{\alpha }-\eta^{\alpha }\right)}^{2}}\;,$$
Then, the global wave function at the ground state will be
(73)$$\psi_{0}\left(\eta^{1},\ldots ,\eta^{m}\right)=\prod_{\alpha =1}^{m}\psi_{\alpha ,0}\left(\eta^{\alpha }\right)\;,$$
with $\lambda =\sum_{\alpha =1}^{m}\frac{1}{n}=\frac{m}{n}$, which is the intrinsic Cramér–Rao lower bound (ICRLB) for *m* sources of information of *n* samples, with each source being modeled with an informative parameter μ, i.e., a total of *m* informative parameters. For further details, see [10]. The global probability density function at the ground state can be written as
(74a)ρ0(η1,…,ηm)=∥ψ0(η1,…,ηm)∥2=∏α=1mψα,02(ηα),(74b)=n2πm2∏α=1mγα2−12exp−n2∑α=1m1γα2y¯α−ηα2,(74c)=n2πm2det(D)−12exp(−n2y¯α−ηαTD−1y¯α−ηα).
Since $\eta =U^{T}\mu$ and $\bar{y}=U^{T}\bar{x}$, where U is an orthonormal m×m matrix and, therefore, |detU|=1, we can express Equation (Equation 74) as a function of the original coordinates
(75)$$\rho_{0}\left(\mu^{1},\ldots ,\mu^{m}\right)={\left(\frac{n}{2\pi }\right)}^{\frac{m}{2}}{\left(\det(\Sigma_{0})\right)}^{-\frac{1}{2}}\exp\left(-\frac{n}{2}\;{\left(\bar{x}-\mu \right)}^{T}\Sigma_{0}^{-1}\left(\bar{x}-\mu \right)\right)$$
However, there are many other solutions in (Equation 69) considering different values of ν in (Equation 71). It is well-known that the solutions of the quantum harmonic oscillator involve Hermite Polynomials, which were introduced elsewhere [8,9]. In particular, for ν=1, we will have $\lambda_{\alpha ,1}=\frac{3}{n}$ and the wave function at the first-excited state will be
(76)$$\psi_{\alpha ,1}\left(\eta^{\alpha }\right)={\left(\frac{n^{3}}{2\pi \gamma_{\alpha }^{2}}\right)}^{\frac{1}{4}}\left(\frac{{\bar{y}}^{\alpha }-\eta^{\alpha }}{\gamma_{\alpha }}\right)\;e^{-\frac{1}{4\gamma_{\alpha }^{2}}n{\left({\bar{y}}^{\alpha }-\eta^{\alpha }\right)}^{2}}\;.$$
We can obtain other solutions via the Hermite polynomials, representing excited states of the quantum harmonic oscillator. For instance, we may obtain the solution $\psi_{\alpha ,\nu }$ for each of the sources α=1,…,m and for each energy level ν=0,…,k. Combining the *m* sources with the k+1 energy levels, we can build up all possible solutions and, therefore, obtain up to ${\left(k+1\right)}^{m}$ different solutions
(77)$$\psi \left(\eta^{1},\ldots ,\eta^{m}\right)=\prod_{\alpha =1}^{m}\psi_{\alpha ,\epsilon_{\alpha }}\left(\eta^{\alpha }\right)\;,$$
where $\epsilon_{\alpha }\in \left\{0,1,\ldots ,k\right\}$ and $\lambda =\sum_{\alpha =1}^{m}\left[\frac{2}{n}\left(\epsilon_{\alpha }+\frac{1}{2}\right)\right]=\frac{m}{n}+\frac{2}{n}\sum_{\alpha =1}^{m}\epsilon_{\alpha }$, the total energy of *m* oscillators, such that $\frac{m}{n}\leq \lambda \leq \frac{2mk}{n}$.

### 2.11. Bayesian Framework and Posterior Probability Density Function

Regardless of having independent or dependent sources of information, we can compute the posterior probability distribution calculated from the sources of information for all data values using Bayes’theorem [13], taking the Riemannian volume of the metric as a prior. This measure is Jeffrey’s prior distribution on the parameter, and it can be considered in some way an *objective*, or at least a *reference* choice for a prior distribution [14].

Considering Equation (Equation 49), the Riemannian volume $\sqrt{\det(G)}$ is constant in Θ. Then, taking into account the likelihood probability distribution (Equation 51), the posterior probability distribution $f_{m,n}\left(\mu ;x^{1},\cdots ,x^{n}\right)$ based on Jeffrey’s prior is equal to
(78a)fm,n(μ;x1,⋯,xn)∝fm,n(x1,⋯,xn;μ)g,(78b)∝exp(−12ndM2(x¯,μ)),(78c)=n2πm2det(Σ0)−12exp(−n2dM2(x¯,μ)),(78d)=n2πm2det(Σ0)−12exp(−n2(x¯−μ)TΣ0−1(x¯−μ))
This posterior probability density function coincides with the global probability density function at the ground state (Equation 75). Precisely, the probability density function of *m* quantum harmonic oscillators at the ground state given by the square of the wave function coincides with the Bayesian posterior obtained from *m* sources of information for all data values using the improper Jeffrey’s prior. This unexpected and exciting result reveals a plausible relationship between energy and Bayes’ theorem.

## 3. Discussion

This paper aimed to expand and refine the mathematical framework initially presented in [1]. We made specific adjustments to the approach, enabling us to consider real-world scenarios more thoroughly. As we continued with our work, we came to appreciate the “intrinsic” nature of the modeling, which we believe is a crucial aspect of our study. Our ultimate objective was to improve upon the foundation established in the previous paper and create an even more robust and accurate framework.

First, we extended the approach by modeling a single source of information with a univariate normal probability distribution $N(\mu ,\sigma_{0})$, as before, but with a constant $\sigma_{0}$ not necessarily equal to 1. We calculated the stationary states in the Riemannian manifold by invoking Schrödinger’s equation to discover that the information could be broken into quantum harmonic oscillators as before but with the energy levels being independent of $\sigma_{0}$, an unexpected but relevant result that motivated us to continue exploring this field.

This primitive result led us to title the work “Intrinsic information-theoretic models”, which asserts that the critical features of our modeling process, such as the energy levels, remain independent of the parametrization used and invariant under coordinate changes. This notion of invariance is significant because it implies that the same model can be applied across different parameterizations, allowing for greater consistency and generalizability. Furthermore, this approach can lead to a more robust and reliable modeling process, as it reduces the impact of specific parameter choices on the final model output. As such, the notion of “intrinsic” information-theoretic models has the potential to improve modeling accuracy and reliability significantly.

Similar to our previous study [1], we evaluated the performance of the estimation of the parameter μ. Instead of calculating the estimator’s variance, we used the expectation of the quadratic Mahalanobis distance to the sample mean to discover that equals the energy levels of the quantum harmonic oscillator, being the minimum quadratic Mahalanobis distance at the minimum energy level of the oscillator. Interestingly, we demonstrated that quantum harmonic oscillators reach the “intrinsic” Cramér–Rao lower bound on the quadratic Mahalanobis distance at the lowest energy level.

Then, we modeled *m* independent sources of information and computed the global density function at the ground state as an example. Essentially, we modeled sources with a multivariate normal probability distribution $N_{m}\left(\mu ,\Sigma_{0}\right)$, with a variance–covariance matrix $\Sigma_{0}$ different than the identity matrix of *m*-dimension, $I_{m}$, but being diagonal initially to describe the independence of the sources of information.

We advanced the mathematical approach by modeling *m* dependent sources of information with a variance–covariance matrix $\Sigma_{0}$ not necessarily diagonal, depicting dependent sources. This resulted in Schrödinger’s equation of *m*-coupled quantum harmonic oscillators. We could effectively decouple the oscillators through a coordinated transformation, thereby partitioning the information into independent modes. This enabled us to obtain the same energy levels, albeit now with respect to the modes, which further proves the “intrinsic” property of the mathematical framework.

Finally, as in our previous study, we showed that the global probability density function of a set of *m* quantum harmonic oscillators at the lowest energy level, calculated as the square modulus of the global wave function at the ground state, equals the posterior probability distribution calculated using Bayes’ theorem from the *m* sources of information for all data values, taking as a prior the Riemannian volume of the informative metric. This is true regardless of whether the sources are independent or dependent.

Apart from the mathematical discoveries detailed in this paper, this framework offers multiple alternatives that we are currently exploring. For example, the informational representation of statistical manifolds with $\Sigma_{0}$ is unknown. Also, this approach can be generalized by exploring other statistical manifolds and depicting how physical observables such as space and time may emerge from linear and nonlinear transformations of a set of parameters of a specific statistical manifold. This way, the laws of physics, including time’s arrow, will appear afterward.

Moreover, several fascinating inquiries warrant further investigation. These involve delving into the relationship between energy and information already highlighted in our initial work. In addition, the very plausible connection between energy and Bayes’ theorem also deserves further exploration. By delving deeper into these topics, we may unlock even more insights into the universe’s fundamental nature and mathematical laws.

The updated framework presented in this study offers a more realistic approach by allowing the modeling of *m*-dependent sources. In real-world scenarios, information is often distributed over multiple sources that may not be entirely independent. By formulating the problem in terms of modes, we can obtain a solution or set of solutions for the proposed framework. This approach provides a valuable tool for solving complex problems that require a deeper understanding of the underlying dynamics.

## Data Availability

No new data were created.

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
