# Peer review of "Intrinsic Information-Theoretic Models"

_entropy, 2024, doi:10.3390/e26050370_

Round 1

Reviewer 1 Report

Comments and Suggestions for Authors

Please see the attachement for the report.

Comments on the Quality of English Language

No

Author Response

Please, attached you will find our reply.

Reviewer 2 Report

Comments and Suggestions for Authors

Author Response

(The authors gave the same response as above.)

Reviewer 3 Report

Comments and Suggestions for Authors

This interesting paper expands and refines  the mathematical structures  presented in Reference

in [1]. 

Specific optimizations enabled for  a better and extended approach with the ultimate objective of giving a firmer  foundation to the formalismes  of [1].

Stationary states were dealt in the Riemannian manifold by invoking Schrodinger’s equation to discover that the ensuing  information could be broken into quantum harmonic oscillators in a better way than in [1]. 

The critical features of their modeling process remain independent of the parametrization used and invariant under coordinate changes. 

The same model can be applied across different parameterizations, allowing for greater consistency and generalizability. 

Quantum harmonic oscillators reach the "intrinsic" Cramer–Rao lower bound on the quadratic Mahalanobis distance at the lowest energy level.  

As in their previous study [1] they showed that the global probability density function of a set of m quantum harmonic oscillators at the lowest energy level, calculated as the square modulus of the global wave function at the ground state, equals the posterior probability distribution calculated using Bayes’ theorem from the m sources of information for all data values, taking as a prior the Riemannian volume of the informative metric. 

The paper is well written and didactic.

I liked it and suggest acceptance.

Author Response

(The authors gave the same response as above.)
